# Alterations in Genes *rib*, *scpB* and Pilus Island Decrease the Prevalence of Predominant Serotype V, Not III and VI, of *Streptococcus agalactiae* from 2008 to 2012

**DOI:** 10.3390/pathogens11101145

**Published:** 2022-10-03

**Authors:** I-An Tsai, Yaochi Su, Ying-Hsiang Wang, Chishih Chu

**Affiliations:** 1Ph.D. Program of Agriculture Science, College of Agriculture, National Chiayi University, Chiayi City 600, Taiwan; 2Department of Veterinary Medicine, College of Veterinary Medicine, National Chiayi University, Chiayi City 600, Taiwan; 3Department of Pediatrics, Chang Gung Memorial Hospital, Puzi City 613, Taiwan; 4Department of Microbiology, Immunology and Biopharmaceuticals, College of Life Sciences, National Chiayi University, Chiayi City 600, Taiwan

**Keywords:** *Streptococcus agalactiae*, antimicrobial susceptibility, pilus island genes, virulence genes, *scpB* gene, *rib* gene

## Abstract

*Streptococcus agalactiae* (GBS) can infect newborns, pregnant women and immunocompromised or elderly people. This study aimed to investigate differences in three pilus genes and virulence genes *pavA*, *cfb*, *rib* and *scpB* and changes in predominant serotypes III, V and VI from 2008 to 2012. The susceptibilities to penicillin, ceftriaxone, azithromycin, erythromycin, clindamycin, levofloxacin and moxifloxacin of 145 GBS strains of serotype III, V and VI strains from 2008 and 2012 were determined using disc diffusion method. PCR identification of ST-17, the pilus genes and virulence genes; multilocus sequence typing (MLST); and conserved domain and phylogenetic analysis of *scpB-1* and *scpB-2* proteins were performed. A dramatic number reduction was observed in serotype V, not III and V, from 2008 to 2012. The rate of resistance to azithromycin, clindamycin and erythromycin was the highest in serotype V. ST-17 was only found in serotype III with pilus genes PI-1+PI-2b. The major pilus genotype was PI-1+PI-2a. Serotype V without the *rib* gene was reduced in number between two studied years. Compared to *scpB-1*, *scpB-2* had a 128-bp deletion in a PA C5a-like peptidase domain and putative integrin-binding motif RGD. In conclusion, reduction in serotype V may be due to presence of *scpB-2* or lack of genes *scpB* and *rib*.

## 1. Introduction

*Streptococcus agalactiae* (group B *streptococcus*; GBS) is a major cause of early-onset or late-onset neonatal sepsis or meningitis and infections in pregnant women and non-pregnant adults, especially those with underlying diseases such as diabetes and in immunocompromised or elderly people [1,2]. Penicillin is recommended to treat perinatal GBS infection [3]; nevertheless, erythromycin and clindamycin are alternative drugs to treat patients that are allergic to penicillin. Therefore, erythromycin- or clindamycin-resistant GBS has increased recently [4,5].

Several methods have been developed to detect GBS infection in humans and animals and to differentiate GBS strains. Ten GBS serotypes can be differentiated by multiplex PCR targeting capsular polysaccharide genes [6]. Furthermore, multiple locus sequence typing (MLST) can be used to determine the sequence patterns (ST) and clonal complexes (CCs), that can be used to trace the disease outbreak and determine the phylogenetic tree [7]. GBS ST-17 strains are highly invasive and virulent for neonatal infections and can be found mainly in serotype III and rarely in serotypes Ia and VI [8]. Therefore, rapid PCR identification of ST-17 has been developed using the *gbs2018* gene as a target [9].

Several virulence factors are involved in GBS infections, such as pilus for GBS attachment on mucosal epithelial cells. GBS carries at least one of the two pilus island genes PI-1 and PI-2, and the latter is categorized into PI-2a and PI-2b [10]. The major highly virulent CC17 strains contain PI-1 and PI-2 genes, and such an allele combination is rarely seen in other clonal complexes [11]. Furthermore, other virulence genes for GBS pathogenesis are the *rib* gene for surface protein [12], the *pavA* gene to regulate fibrinogen-binding protein for human cell adhesion [13], the *scpB* gene for the C5a peptidase that inhibits neutrophil recruitment [14], the and *cfb* gene for the pore toxin diffusible, thermostable, extracellular protein (CAMP) factor [15].

Genomic variations may have contributed to changing the predominant serotypes from Ib and V to III and VI between 2008 and 2012 [1]. Other factors may be involved in the change in predominant serotypes. Therefore, our aim was to investigate differences in these virulence genes for change in the predominant serotypes III, V and VI from 2008 to 2012.

## 2. Results

### 2.1. Antimicrobial Susceptibility Analysis

The number of strains differed among serotypes with little increase for serotypes III and VI and a dramatic decrease in serotype V (Table 1). All strains were sensitive to penicillin, ceftriaxone, levofloxacin and moxifloxacin, and 53.1% of the strains were resistant at least to one of three antimicrobials azithromycin, clindamycin and erythromycin. The resistance rate was 51% for azithromycin, 49% for clindamycin and 49.7% for erythromycin overall and differed among three serotypes, being the highest in serotype V and the lowest in serotype VI. Most resistant strains were resistant to all three antimicrobials but differed in prevalence between the two periods with an increase from 29.4% to 38.9% in serotype III, a decrease from 60.3% to 53.3% in serotype V and a decrease from 41.2% to 25% in serotype VI. 

### 2.2. ST-17 and Pilus Island Gene Analysis

GBS ST-17 strains were detected by polymerase chain reaction (PCR) and were only present in serotype III, with rates of 35.3% and 33.3% in the two years, respectively (Figure 1A and Table 2). These ST-17-positive strains were confirmed with MLST analysis with the allelic profile of 2112111. Furthermore, ST-17 was not found in serotypes V, VI and Ia. All GBS strains contained at least one of the pilus genes PI-1, PI-2a and PI-2b, with PI-2a being the most prevalent (87.6%), followed by PI-1 (77.9%) and PI-2b (12.4%) (Figure 1B and Table 2). Four pilus gene profiles were PI-1+PI-2a (71%), PI-1+PI-2b (6.9%), PI-2a (16.6%) and PI-2b (4.1%). The prevalence of PI-1+PI-2a decreased from 76.1% to 62.3% between 2008 and 2012, with a dramatic decrease in prevalence from 94.1% (16/17) to 60% (12/20) in serotype VI, and in number from 43 (74.1%) to 11 (73.3%) in serotype V. The prevalence of PI-2a increased from 12% to 24.5% on average between the two studied years and differed among serotypes III, V and VI. All serotype III/ST-17 strains carried PI-2b with PI-1+PI-2b as the majority; however, PI-2b was also found in serotype III in 2012 and serotype V in 2008 only.

### 2.3. Virulence Gene Analysis

Four virulence genes, namely *pavA*, *cfb*, *scpB* and *rib,* in *S. agalactiae* were detected with different patterns: one PCR product for *pavA* and *cfb*; three PCR products for *scpB*, namely 567-bp *scpB-1*, 439-bp *scpB-2* and no PCR product; and two PCR products for *rib,* namely 369-bp PCR product and no PCR product (Figure 2). The prevalence of these genes was 100% for both *pavA* and *cfb*, followed by 97.2% for *scpB* and only 34.5% for *rib* (Table 3). In *scpB,* more strains carried *scpB-1* (60%) than *scpB-2* (37.2%), which was only observed in serotype V with number reduction from 2008 to 2012, and no *scpB* was found in serotype V in 2008, not 2012. All serotype III and VI strains carried the *scpB-1*. An increase in the *rib* gene was observed from 21.7% to 56.6% between the two periods, due to a reduction in no-*rib* strains in serotype V from 2008 to 2012. Furthermore, the prevalence of *rib* differed among the three serotypes, being more than 88% in serotype III in both periods and exhibiting a decrease and an increase between the two periods in serotypes V and VI, respectively.

### 2.4. Amino Acid Comparison and Conserved Domain (CDD) Analysis of scpB Protein

The genes *scpB-1* and *scpB-2* of three serotypes were sequenced and compared with sequences of accession numbers AF189002.2 (*scpB*-*1*) and FJ752116 (*scpB-2*). The 567-bp *scpB-1* was amplified at nucleotide positions from 969 to 1535, and 439-bp *scpB-2* showed a 128-bp deletion at nucleotide positions from 1058 to 1185 and at amino acid sequences from 353 to 395 (Figure 3). CDD analysis of *scpB* protein found a protease-associated domain containing proteins like *Streptococcus pyogenes* C5a peptidase domain (PA C5a-like peptidase domain) at amino acid positions 346-475 and putative integrin binding motif RGD at amino acid positions 389-391. The *scpB-2* gene lacked this motif and partial peptidase domain. Using AF189002.2 as a standard, mutations were determined at L352I and Q447P in serotype III and at D342A, H344Q, N400K and R437K in serotype V (*scpB-2* gene) and in serotype VI G776 strain *(scpB-1* gene), and no mutations were determined in strains of serotypes V and VI (*scpB-1* gene). 

### 2.5. Phylogenetic Analysis

The phylogenetic analysis of amino acid sequences separated all strains into two groups, A (*scpB-1*) and B (*scpB-2*, except strain G776) (Figure 4). Group A was categorized into two subgroups, A1 (serotype III) and A2 (serotypes V and VI). Furthermore, group B was separated into subgroup B1 (serotype V with five mutations) and subgroup B2 (serotypes V and VI with four mutations). Interestingly, the *scpB-1* gene in the serotype VI G776 strain had mutations identical to those in the *scpB-2* gene, and another serotype VI G260 had no mutations in *scpB-1*.

## 3. Discussion

Group B *Streptococcus* (GBS) serotypes evolve differently. Earlier, we reported the change of predominant serotypes from Ib and V to III and VI due to a difference in genomic variation with a dramatic decrease in serotype V, not in serotypes III and VI (Table 1) [1]. Other factors may be involved in such a change in predominant serotypes. Antibiotic susceptibility analysis demonstrated that these three serotypes differed in their rate of resistance to clindamycin, erythromycin and azithromycin; further, the rate of resistance to clindamycin and erythromycin was higher than that of isolates from studies conducted in Switzerland and Korea (except serotype V) (Table 4) [16,17]. The overall rate of resistance to clindamycin and erythromycin found in this study was lower than that found in studies in China [8,18] and the strains in the USA [5] and higher than that in Brazil and Iran [19,20]. As mentioned above, the highest antimicrobial resistance rate was not associated with number reduction in serotype V, suggesting other factors may be involved.

Highly invasive GBS serotype III/ST-17 is the main cause of neonatal sepsis or meningitis in China, France and Serbia [21,22,23]. In this study, we confirmed that PCR identification of ST-17 is a useful method and that ST-17 was only found in serotype III, not in serotypes Ia and VI (Table 2); these results differed from those in a previous report of the existence of ST-17 in serotypes Ia and VI in China [8]. Pili play an important role in bacterial attachment to epithelial cells. GBS contains three pilus genes, namely PI-1, PI-2a and PI-2b [10]. PI-1+PI-2a is the major pilus type in China, Iran and Taiwan (Table 2) [18,20] and is responsible for colonization and biofilm formation [24]. However, serotype III/ST-17 showed two pilus types, PI-1+PI-2b and PI-2b, with PI-2b being the predominant (Table 2). These results confirm PI-2b in highly invasive ST-17 strains, which is important for GBS and host interaction during GBS infection [11]. Furthermore, PI-2b can be found in serotypes Ia and II [20], serotype V (Table 2) and serotype VI [25]. In this study, serotype V with PI-2b was observed in 2008 and disappeared in 2012 (Table 2), suggesting that PI-2b is not important for adherence or invasion of serotype V and that other virulence genes are involved in the reduction in serotype V.

All serotype III/ST-17 strains had four virulence genes, *pavA*, *cfb*, *rib* and *scpB*. Despite the presence of virulence genes *pavA* and *cfb* in all strains, the prevalence of *rib* and *scpB* had no change in serotype III and increased in serotypes V and VI between the two studied years (Table 3). As a member of the Alp protein family with highly repetitive sequences stimulating protective immunity, the *rib* gene was identified in 69.8% of GBS isolates and was associated with serotypes II, III and V, especially in the serotype III/ST-17 strains [8,12,26]. In Romania, the *rib* gene was found in 73.7% of serotype III strains but was absent in serotype V strains [27], while in China, the *rib* gene was found in 70.6% of serotype III strains and only in 10.5% of serotype V strains [8]. These results demonstrate the serotype association of the *rib* gene. In this study, the *rib* gene was found in 88.5% of serotype III strains, 21.6% of serotype VI strains and 15% of serotype V strains, and the prevalence of the *rib* gene between the two periods did not change in serotype III and increased in serotype V and VI (Table 3). However, a significant reduction in number from 55 to 7 was found in serotype V from 2008 to 2012. These results imply that the *rib* gene is an important virulence factor. 

CDD analysis determined that the *scpB* gene encodes a surface-localized serine protease: C5a-like peptidase, which inactivates human C5a to inhibit complement activation, with a putative integrin-binding motif RGD (Figure 3) [26]. Commonly, GBS strains carry the *scpB* gene [28,29,30]. In the present study, we identified two *scpB* genes: full-length *scpB-1* and deletion *scpB-2* (Figure 3). *scpB-2* was reported in the USA in 2009 [31] and lacked the partial C5a-like peptidase and putative integrin-binding motif RGD (Figure 3). Further, *scpB-2* gene was only found in serotype V; a change in number from 42 to 5 was found for strains with the *scpB-2* gene, and an increase from 2 to 5 for those strains with the *scpB-2* and *rib* genes in serotype V (Table 3). These data suggest that genes *scpB-1* and *rib* are important virulence factors. The amino acid comparison and phylogenetic analysis of *scpB genes* demonstrated serotype-associated mutations and the evolution of *scpB-2* in serotype V possibly from *scpB-1* in serotype VI (Figure 3 and 4). This phenomenon may be due to phase variations. 

## 4. Materials and Methods

### 4.1. Bacterial Isolates

In total, 145 GBS isolates of serotypes III, V and VI were collected from Chiayi Chang Gung Memorial Hospital in 2008 and 2012 [1]. The GBS genomic DNA was purified using a DNA extraction kit (Quality Systems, Taipei, Taiwan). Purified DNA was separated in 0.8% agarose gel in 0.5× TBE buffer at 100 V for 35 min. After staining with ethidium bromide, images were recorded, analyzed and quantified using ImageJ free software (https://imagej.nih.gov/ij/; Accessed on 03 June 2022).

### 4.2. Antimicrobial Susceptibility

Susceptibility to penicillin, ceftriaxone, azithromycin, erythromycin, clindamycin, levofloxacin and moxifloxacin was determined by disc diffusion method according to guidelines of the Clinical and Laboratory Standards Institute (CLSI) [32]. *Streptococcus pneumoniae* ATCC49619 was used as standard.

### 4.3. PCR Identification of ST17 Strains

PCR amplification of ST-17 was performed as previously described with the primers ST-17S and ST-17AS (Table 5) [9]. PCR products were separated in 1.5% agarose gel in 0.5% TBE buffer at 100 V for 35 min. After staining with ethidium bromide, images were recorded and analyzed. Furthermore, MLST analysis was performed for PCR ST-17-positive isolates by sequencing the seven housekeeping genes (*adhP, pheS, atr, glnA, sdhA, glcK* and *tkt*) [7]. GBS sequence types were determined using the *S. agalactiae* MLST database (http://pubmlst.org/sagalactiae/; Accessed on 03 June 2022).

### 4.4. Pilus Island Genes, Virulence Genes and Sequencing

The pilus island genes PI-1, PI-2a and PI-2b were identified by multiplex PCR with the primer sets listed in Table 5. PI-1-negative isolates were confirmed further by additional PCR reaction using primers PI1_all [33]. Virulence genes *rib*, *pavA*, *scpB* and *cfb* were identified by previous PCR methods with the primers listed in Table 5 [30]. PCR products were separated in 1.5% agarose gel in 0.5× TBE buffer at 100 V for 35 min. After staining with ethidium bromide, images were recorded and analyzed. We randomly selected different PCR products of *scpB-1* and *scpB-2* amplified from three serotypes for sequencing. The forward and reverse sequences were assembled with the SeqMan software of Lasergene DNAStar 7.1. Assembled sequences were annotated using the NCBI BLAST program (https://blast.ncbi.nlm.nih.gov/Blast.cgi; Accessed on 10 March 2022).

### 4.5. Amino Acid Comparison and Conserved Domain Analysis of scpB Protein

The *scpB* protein was analyzed with 8 strains, which were serotype III (G311), serotype V (G266 and G369) and serotype VI (G260) from 2008 and serotype III (G765), serotype V (G690 and G739) and serotype VI (G776) from 2012. Accession sequence AF189002.2 was used as standard [34]. Sequence alignment was performed using the MegAlign software of Lasergene DNAStar 7.1. The conserved domain of the *scpB* protein was analyzed using the Conserved Domains Database (https://www.ncbi.nlm.nih.gov/cdd/; 10 April 2022). 

### 4.6. Phylogenetic Analysis of scpB Gene

The amino acid sequences of 8 strains were aligned with those sequences of 19 strains from 10 countries including the United States, Canada, Japan, China and Singapore from the NCBI Genbank database. Phylogenetic analysis of *scp**B* protein was performed using the software Molecular Evolutionary Genetic Analysis (MEGA 11, https://megasoftware.net, 12 May 2022). The phylogenetic tree was constructed using (UPGMA) with bootstrap tests (1000 replicates). The branch lengths of this tree are the same units used to infer evolutionary distances in phylogenetic trees. Evolutionary distances were calculated using the *p*-distance method with 7 threads of system resource usage. All ambiguous positions for each sequence pair were deleted (pairwise deletion option). 

## 5. Conclusions

In conclusion, the predominant change in three serotypes between two periods may be due to the presence of ST-17 in serotype III with PI-1+PI-2b, *scpB-1* and *rib* genes in serotypes III and VI. The *scpB-1* protein contains a C5a-like peptidase and lacks this domain in *scpB-2*. Therefore, either the lack of the *scpB* gene and *rib* gene or the presence of *scpB-2* may have some influence on the reduction in the number of serotype V strains. 

## Figures and Tables

**Figure 1 pathogens-11-01145-f001:**
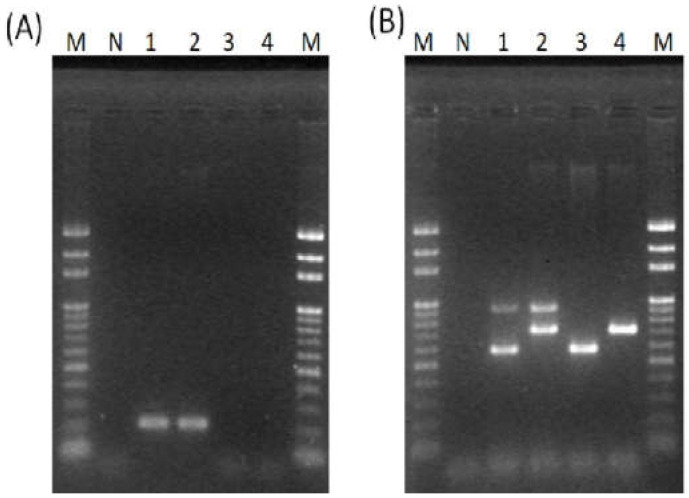
Gel electrophoresis of PCR products for ST-17 (**A**) and pilus island genes (**B**). M: 100 bp DNA ladder. N: negative control. The size of PCR product was 210 bp for ST-17, 886 bp for PI-1, 575 bp for PI-2a and 721 bp for PI-2b.

**Figure 2 pathogens-11-01145-f002:**
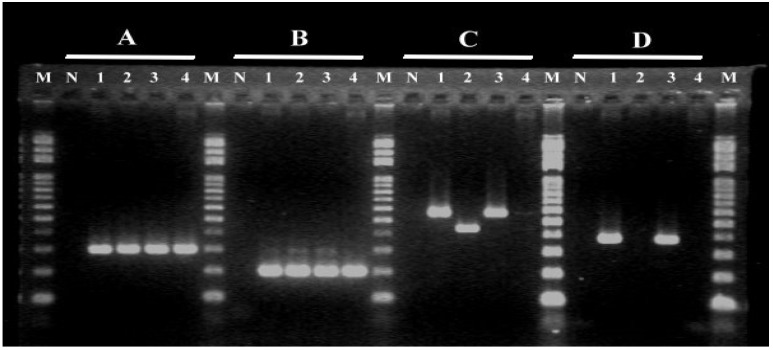
Gel electrophoresis of PCR products for *pavA* gene (288 bp) (**A**), *cfb* gene (193 bp) (**B**), *scpB* gene (567 bp or 439 bp) (**C**) and *rib* gene (369 bp) (**D**). M: 100 bp DNA ladder. N: negative control. Lines 1-4 are GBS strains G311, G369, G757 and G788.

**Figure 3 pathogens-11-01145-f003:**
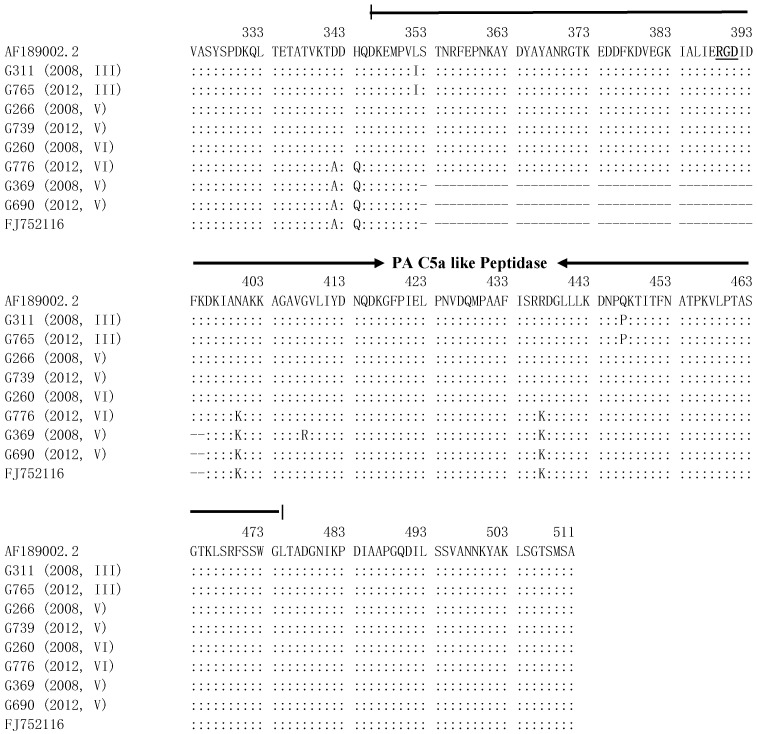
Amino acid sequence alignment from amino acids 324 to 511 of *scpB* gene. Amino acid positions 346 to 475 are a PA C5a-like peptidase domain. RGD is a putative integrin-binding motif. Colons represent the identical amino acid and horizontal lines represent a deletion.

**Figure 4 pathogens-11-01145-f004:**
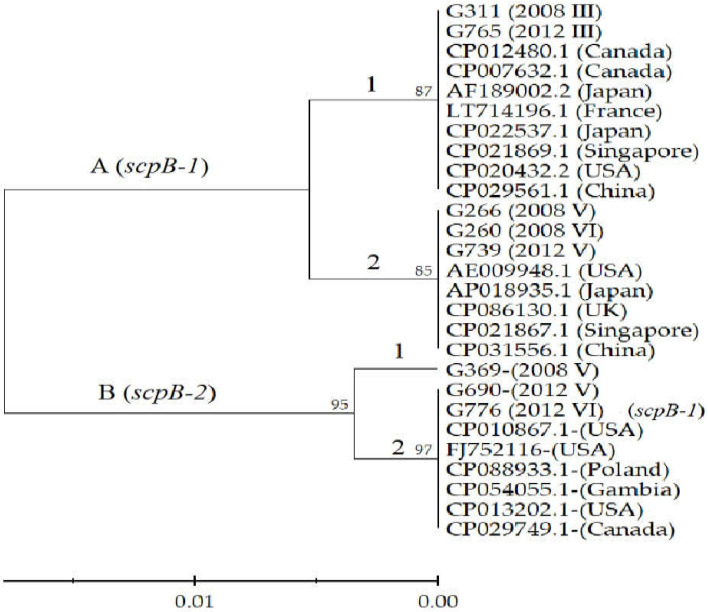
Phylogenetic tree of the *scpB* protein of 8 strains in this study and 19 strains from Genbank. The phylogenetic tree was constructed using the unweighted pair group method with arithmetic mean (UPGMA) with bootstrap tests (1000 replicates) shown above the branches. Evolutionary distances were calculated using the p-distance method, in units of amino acid differences per site. All ambiguous positions for each sequence pair were deleted (pairwise deletion option).

**Table 1 pathogens-11-01145-t001:** Rate of resistance to three drugs in serotype III, V and VI strains isolated from 2008 and 2012.

Number of Antimicrobial Resistances	Antimicrobial Agents	Serotype III (%)		Serotype V (%)		Serotype VI (%)		Total (%)		Sum (%)
		2008	2012	2008	2012	2008	2012	2008	2012	
0	-	8 (47.1)	8 (44.4)	21 (36.2)	7 (46.7)	10 (58.8)	14 (70)	39 (42.4)	29 (54.7)	68 (46.9)
1	AZM			1 (1.7)				1 (1.1)	0	1 (0.7)
	CC	1 (5.9)	2 (11.1)					1 (1.1)	2 (3.8)	3 (2.1)
2	E AZM	3 (17.6)	1 (5.6)				1 (5)	3 (3.3)	2 (3.8)	5 (3.4)
	AZM CC			1 (1.7)				1 (1.1)	0	1 (0.7)
3	E AZM CC	5 (29.4)	7 (38.9)	35 (60.3)	8 (53.3)	7 (41.2)	5 (25)	47 (51.1)	20 (37.7)	67 (46.2)
Total		17 (18.5)	18 (34)	58 (63)	15 (28.3)	17 (18.5)	20 (37.7)	92	53	145

AZM = azithromycin; CC = clindamycin; E = erythromycin.

**Table 2 pathogens-11-01145-t002:** PCR detection of ST-17 and pilus island genes in serotype III, V and VI strains isolated from 2008 and 2012.

Total (N (%))	ST-17 (PCR/MLST)	PI gene (N (%))			Serotype III (N (%))		Serotype V (N (%))		Serotype VI (N (%))	
		PI-1	PI-2a	PI-2b	2008	2012	2008	2012	2008	2012
10 (6.9)	+/+	+		+	6 (35.3)	4 (22.2)				
2 (1.4)				+		2 (11.1)				
103 (71)	−/−	+	+		11 (64.7)	10 (55.6)	43 (74.1)	11 (73.3)	16 (94.1)	12 (60)
24 (16.6)			+			1 (5.6)	10 (17.2)	4 (26.7)	1 (5.9)	8 (40)
6 (4.1)				+		1 (5.6)	5 (8.6)			
145	12 (8.3)	113 (77.9)	127 (87.6)	18 (12.4)	17	18	58	15	17	20

+: presnece; −: absence.

**Table 3 pathogens-11-01145-t003:** Presence of virulence genes *pavA*, *cfb*, *scpB* and *rib* in serotype III, V and VI strains isolated from 2008 and 2012.

Total N (%)	Virulence gene (N (%))					Serotype III (N (%))		Serotype V (N (%))		Serotype VI (N (%))	
	*pavA*	*cfb*	*scpB-1*	*scpB-2*	*rib*	2008	2012	2008	2012	2008	2012
43 (29.7)	+	+	+		+	15 (88.2)	16 (88.9)	1 (1.7)	3 (20)	2 (11.8)	6 (30)
7 (4.8)	+	+		+	+			2 (3.4)	5 (33.3)		
44 (30.3)	+	+	+			2 (11.8)	2 (11.1)	9 (15.5)	2 (13.3)	15 (88.2)	14 (70)
47 (32.4)	+	+		+				42 (72.4)	5 (33.3)		
4 (2.8)	+	+						4 (6.9)			
145	145 (100)	145 (100)	87 (60)	54 (37.2)	50 (34.5)	17	18	58	15	17	20

**Table 4 pathogens-11-01145-t004:** Rate of resistance (%) to three antibiotics in different countries.

Country	Taiwan (this study)				China	China	USA	Brazil	Switzerland			Iran	South Korea			
Year	2008 + 2012				2015	2017–2019	2008–2016	2008–2018	2009–2010			2015–2016	1990–2002			
Serotype	All	III	V	VI	All	All	All	All	III	V	VI	All	All	III	V	VI
Azithromycin	51	45.7	61.6	35.1												
Clindamycin	49	42.9	60.3	32.4	66.4	66.7	43.2	2–16.7	15	24.7	0	21.1	35	29	85	5
Erythromycin	49.7	45.7	58.9	35.1	72.1	70.3	54.8	4–14	15	25.8	0	14	30	23	85	21
Reference					[8]	[18]	[5]	[19]	[16]			[20]	[17]			

**Table 5 pathogens-11-01145-t005:** All primers used in this study.

Gene	Primers	Sequences (5′ to 3′)	Product Size (bp)	Reference
ST-17	ST-17S	ATACAAATTCTGCTGACTACCG	210	[9]
	ST-17AS	TTAAATCCTTCCTGACCATTCC		
*adhP*	adhP-F	GTTGGTCATGGTGAAGCACT	672	[7]
	adhP-R	ACTGTACCTCCAGCACGAAC		
*pheS*	pheS-F	GATTAAGGAGTAGTGGCACG	723	
	pheS-R	TTGAGATCGCCCATTGAAAT		
*atr*	atr-F	CGATTCTCTCAGCTTTGTTA	627	
	atr-R	AAGAAATCTCTTGTGCGGAT		
*glnA*	glnA-F	CCGGCTACAGATGAACAATT	589	
	glnA-R	CTGATAATTGCCATTCCACG		
*sdhA*	sdhA-F	AGAGCAAGCTAATAGCCAAC	646	
	sdhA-R	ATATCAGCAGCAACAAGTGC		
*glcK*	glcK-F	CTCGGAGGAACGACCATTAA	607	
	glcK-R	CTTGTAACAGTATCACCGTT		
*tkt*	tkt-F	CCAGGCTTTGATTTAGTTGA	859	
	tkt-R	AATAGCTTGTTGGCTTGAAA		
PI-1	PI1-UP	GGTCGTCGATGCTCTGGATTC	881	[33]
	PI1-DN	GTTGCCCAGTAACAGCTTCTCC		
PI-2a	PI2a-UP	CTATGACACTAATGGTAGAAC	575	
	PI2a-DN	CACCTGCAATAGACATCATAG		
PI-2b	PI2b-UP	ACACGACTATGCCTCCTCATG	721	
	PI2b-DN	TCTCCTACTGGAATAATGACAG		
PI1_all	PI1_all-UP	ACCTATGTTGCTGATTCGGCTGAAAATG	684 *	
	PI1_all-DN	TACGGACACTTTCTAGTGCCTTTGGATC		
*pavA*	*pavA*-F	TTCCCATGATTTCAACAACAAG	288	[30]
	*pavA*-R	AACCTTTTGACCATGAATTGGTA		
*cfb*	*cfb*-F	ATGGGATTTGGGATAACTAAGCTAG	193	
	*cfb*-R	AGCGTGTATTCCAGATTTCCTTAT		
*scpB*	*scpB*-F	AGTTGCTTCTTACAGCCCAGA	567	
	*scpB*-R	GGCGCAGACATACTAGTTCCA		
*rib*	*rib*-F	CAGGAAGTGCTGTTACGTTAAAC	369	
	*rib*-R	CGTCCCATTTAGGGTCTTCC		

* size of the isolate lacks the PI-1 islet (if this locus is present, a 16.7-kb fragment would be expected).

## Data Availability

Not applicable.

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
