# Peer review of "Alterations in Genes rib, scpB and Pilus Island Decrease the Prevalence of Predominant Serotype V, Not III and VI, of Streptococcus agalactiae from 2008 to 2012"

_pathogens, 2022, doi:10.3390/pathogens11101145_

Round 1

Reviewer 1 Report

The work is well presented but there are some litle revisions required:

Page 3

Figure 1 - please specify in the legend what the numbers 1 , 2, 3 , 4 refers to. as was done in figure 2.

Page 6 :

Table 4 needs reformatting,  as there are words cut

Page 9, line 255 :

Conclusions: There is a very strong afirmative . I think is better to  change

for something like  "either lack of scpB gene and rib gene or presence of scpB-2 may have some influence  in the  reduction 255 in number serotype V strains.

Author Response

Reviewer 1

Comments and Suggestions for Authors

The work is well presented but there are some little revisions required

Page 3 :

Figure 1- Please specify in the legend what the numbers 1, 2, 3, 4 refers to

as was done in Figure 2.

Response

We revised as “Number of antimicrobial resistance” on the top.

Page 6

Table 4 need reformatting, as there are words cut

Response

We rechecked the Table 4 and did not found your suggestion.

Page 9. Line 255

Conclusions: There is a very strong affirmative. I think is better to change.

for something like “either llack of scpB and rib gene or presence of scp-B-2 may have some influence in the reduction in number of serotype V strains.

Response

We changed in text in Line 277-278 as “either lack of scpB gene and rib gene or presence of scpB-2 may have some influence in the reduction in number of serotype V strains.” Based on your recommendation.

We sincerely thanks for your valuable comments to improve our manuscript.

Reviewer 2 Report

Title of article reviewed:  Genes rib, scpB and pilus island responsible for reduction of predominant serotype V, not III and VI, of Streptococcus agalactiae from 2008 to 2012

Dear authors,

You have made a great work of important scientific value.

Your results are successfully analyzed in details. It would be interesting to continue with a new report of currently circulating  strains.

English language needs to be improved, as there are many errors throughout the manuscript.

As well, the title seems a bit confusing may be due to bad wording, so I suggest that the title should be restated.

Analytically, you can find my comments, as they were sent to the editor as well:

SPECIFIC COMMENTS

Title:

I think that the title is a bit confusing. I would suggest to be restated as

 “Decreasing prevalence of predominant S.agalactiae serotype V from 2008 to 2012, due to rib, scpB and pilus island genes alterations. “

Or

“Molecular analysis of rib, scpB and pilus island genes of S.agalactiae strains and their implementation on prevalence of serotypes III, V and VI from 2008 to 2012, in Taiwan. “

I think the second one is more inclusive according to the study and comprehensive.

Abstract:

Line 17: “these virulence genes”

Authors should firstly name the genes and then refer to them as “these”, so the sentence needs rephrasing

Lines 19-20: “ 145 GBS serotypes III, V and VI strains”

Maybe it could be better rephrased as: “ 145 GBS strains of serotypes….”

Lines 21-22: “multilocus sequence analysis (MLST)

First letter of each word must be capitalized.

Line 22: “and conserved domain….”

“and” could be better replaced with “as well as” to be more comprehensive, as a second “and”  follows.

Line 23: There is no verb in the sentence.

“A dramatic number reduction…(what?)…in serotype V…”

Line 25: “genotypes” must be replaced with “genotype”

Line 26:  “rib gene…was…reduced”

“was” must be added

Line 26: “two periods” may be more comprehensive if changed in “two studied years”

It should be changed throughout the whole manuscript.

Introduction

Lines 43-44: First letter of each word must be capitalized for ST, MLST and CC.

Line 46: “strain is a” should be changed in “strains are”

Line 57: ”change” should be better replaced with “contributed to change” or “changed”

Line 59: “aims was” should be replaced with “aim was”

Results

Line 63: May be something is missing here “with little increase….”

May be authors would like to say “with little increase of resistance rate”

Line 61: “the resistance rate…” for which one?

May be authors would like to say “Totally, the resistance rate…”

Line 80: Why do authors refer results among strains of serotype Ia, since they are not included in the study? Maybe they are related to their previous study, but they shouldn’t be included in the analysis of results of the present study.

The same in line 162 (Discussion).

Line 85:  dramatic decrease in number….in serotype V”

I think this is not a finding, as well the whole number of strains of serotype V was reduced. Authors could be better evaluate the results by percentages, where we observe about the same percentage for serotype V strains, while a slight decrease for serotype III and VI strains.

Line 88: “majority however PI-2b…” needs a slight modification in two sentences in order to be better worded, as follows:

“…majority. PI-2b was found in serotype III in 2012 and serotype V, only in 2008.”

Lines 103-105: “In gene scpB……2012”. These sentence needs to be rewritten. It is not clear what authors would like to say. Maybe it is better to make more sentences.

Line 134: “strains”, an “r” is missing

Tables: In all tables the percentages must be under the line of numbers in order to be easier to read.

In all tables legends “three serotypes”  must be better replaced with “serotypes III, V and VI, in 2008 and 2012”

Table 1: Legend: “Resistance rate” than “resistant rate”

Please add the whole number of strains of each serotype in the first line

Table 2: Legend: “genes” the “s” is missing

Figure 3. Legend line 130: “Amino”

Discussion

Line 154: “the isolates from Switzerland”

May be should be better as “isolates from studies conducted in Switzerland”

Line 155: “than the strains…”

May be should be better as “than the respective of studies in China and USA and higher in Brazil and Iran”

Lines 162-63: Please see above the comments for serotype Ia

Line 163: Should be better rephrased “(Table 2). Such results…”

Line 168: “in common” should be changed with “being the predominant”

Line 172: “PI-2b is not important for serotype V”

Is this the clue of the study or we could support that the lack of PI-2b is correlated with the reduction of serotype V? I think this hypothesis is totally the opposite, as it suggests that PI-2b is important for the surviving of serotype V. I would like the authors to make their suggestion more clear here.

Lines 183-184: Should be better rephrased “serotype V strains. Prevalence of rib….had no change in serotype III and increased in…”

Line 185: Should be better rephrased “in number from 55 to 7 was found in serotype V…”

Materials and Methods

Line 203: “Totally”, the “y” is missing also, source of strains should be added.

Line 206: “0.5% TBE” may be “0.5X TBE” It should be changed throughout the manuscript.

Line 212: “pneumoniae”, the “e” is missing

Line 229: What type of sequencing is refered? There are no such results in results paragraph (if it is not my misunderstanding), so why to refer sequencing.

Why do authors refer “randomly selected different PCR products”. Please clarify the method or exclude it at all, if the results are not in the results paragraph.

Line 233: “conserved”

Line 234: Please change to “protein was analyzed within 8 strains….which were of serotype..”

Line 241-242: There is no verb in the sentence. May be authors would like to say “The amino acid sequences were compared/were aligned….with these of 19 strains…”

Line 243: “was performed”

Thank you for choosing me as a reviewer,

Sincerely yours,

A. Xirogianni

Author Response

Reviewer 2

Comments and Suggestions for Authors

Titles

1 I think that the title is a bit confusing. 1 would suggest to be restated as "Decreasing prevalence of predominant S. agalactiae serotype V from 2008 to 2012, due to rib,scpB and pilus island genes alterations. or

Molecular analysis of rib,scpB and pilus island genes of S.agalactiae strains and their implementation on prevalence of serotypes III, V and VI from 2008 to 2012 ,in Taiwan. “

1 think the second one is more inclusive according to the study and   comprehensive.

Response

Thanks for your suggestions. We revise the title as “Alterations in genes rib, scpB and pilus island decrease the prevalence of predominant serotype V, not III and VI, of Streptococcus agalactiae from 2008 to 2012”

Abstract:

Line 17: “these virulence genes"      '、

Authors should.firstly name the genes and then refer to them as

“these", so the sentence needs rephrasing

Response:

We revised in Line 17 as “three pilus genes and virulence genes pavA, cfb, rib and scpB” and in Line 21 as “the pilus genes and virulence genes,”

Lines 19-20: “145 GBS serotypes III,       V and VI strains"

Maybe it could be better rephrased as: 145 GBS strains of serotypes.. .

Response:

We changed accordingly.

Lines 21-22: “multilocus sequence analysis (ML8T)

First letter of each word must be capitalized.

Response:

MLST is an abbreviation needed to be capitalized for original words that are not needed to be captalized for the first letter of the words.  

Line 22: “and conserved domain....

“and" could be better replaced with “as well as" to be more comprehensive, as a second “and"  follows.

Response

Thanks for your recommendations. We change as “and multilocus sequence typing (MLST), as well as conserved domain…”

Line 23: There is no verb in the sentence.

“A dramatic number reduction.. .(what?).. .in serotype V..."

Response

We added “was observed”

Line 25: “genotypes" must be replaced with “genotype"

Line 26:  “rib gene…...was...reduced"

“was" must be added

Line 26: “two periods" may be more comprehensive if changed in “two studied years" It should be changed throughout the whole manuscript.

Response

We changed accordingly.

Introduction

Lines 43-44: First letter of each word must be capitalized for ST, MLST and  CC.

Response

Thanks for your suggestions, However, there in no need to do so.

Line 46: “strain is a" should be changed in “strains are"

Response:

We changed as “strains are highly invasive and virulent…”

Line 57: "change" should be better replaced with “contributed to change" or “changed"

Response

We revised as “be contributed to change"

Line 59: "aims were" should be replaced with “aim was"

Line 66: "the resistance rate." for which one?‘ May be authors:、would like to say ‘Totally, the resistance rate..."

Response

We changed accordingly

Results

Line 63: May be something is missing here “with little increase...." May be authors  would  like to say “with little increase of resistance rate" Response

Response

Thanks for your suggestion. However, in the beginning of the sentence, we mentioned “The number of strains differed”.

Line 80: Why do authors refer results among strains of serotype la, since they are not included in the study?  Maybe they are related to their previous study,but they shouldn't be included in the analysis of results of the present study.

The same in line 162 (Discussion).

Response

Thanks for your recommendation. We did analyze serotype 1a fish and human strains here because other reference reported by researcher in China, which is very close to Taiwan, We want to report as you demonstrated the difference.       

Line 85: “dramatic decrease in number....in serotype V"

1 think this is not a finding, as well the whole number of strains of serotype V was reduced. Authors could be better evaluate the results by percentages, where we observe about the same percentage for serotype V strains, while a slight decrease for serötype III and VI strains.

Response

Thanks for your opinions. The change in prevalence was due to change in total number and the number of specific genotype. . Here, we referred to main change in number for this specific pilus island genotype.

Line 88: majority however PI-2b..." needs a slight modification in two sentences in order to be better worded, as follows:

“...majority. PI-2b was found in serotype III in 2012 and serotype V, only in 2008."

Response

We change accordingly.

Lines 103-105: “In gene scpB ...一 .2012". These sentence needs to be rewritten. It is not clear what authors would like to say. Maybe it is better to make more sentences.

Response

We revised as “The prevalence of these genes was 100% for both genes pavA and cfb; followed by 97.2% for gene scpB, and only 34.5% for gene rib (Table 3).”

Line 134: “strains" ,an “r" is missing

Response

We revised as “and no mutation in strains of serotype V and VI (scpB-1 gene).”

Tables:

In all tables the percentages must be under the line of numbers in order to be easier to read.

In all tables  legends  three  serotypes" must  be  better  replaced with serotypes III, V and VI,in 2008 and 2012"

Response

We 鄒

Table 1: Legend: “Resistance rate" than “resistant rate"

Please add the whole number of strains of each serotype in the first line

Response

We changed as resistance rate. However, we did not move the number in the beginning for evaluation together between serotypes and total number. 

Table 2: Legend: “genes" the “s" is missing

Response

We revised accordingly.

Figure

Figure 3. Legend line 130: “Amino"

Response, We changed accordingly

Discussion

Line 154: "the isolates from Switzerland"

May be should be better as “isolates from studies conducted in Switzerland"

Line 155: than the strains..."

May be should be better as “than the respective of studies  inChina and USA and higher in Brazil and Iran"

Response

We revised accordingly.

Lines 172-73: Please see above the comments for serotype 1

Response

Thanks for your recommendation. We reported as your comments. We want to demonstrate difference in serotype between China and Taiwan.  

Line 173: Should be better rephrased “(Table 2).

Such results..

Response

We revised in text as “such results differed from the previous report of existence of ST-17 in serotype Ia and VI in China [8]”.

Line  178:  “in  common"  should  be  changed  with  “being the predominant"

Response

We revised accordingly.

Line 172: “PI-2b is not important for serotype V"

Is this the clue of the study or we could support that the lack of PI-2b is correlated with the reduction of serotype V? 1 think this hypothesis is totally the opposite, as it suggests that  PI-2b is important for the surviving of serotype V. 1 would like the authors to make their suggestion more clear here.

Response

We revised in text in Line 184-185 as “suggesting that PI-2b is not important for adherence or invasion for serotype V”

Lines 183-184: Should be better rephrased "serotype V strains. Prevalence of ríb....had no change in serotype III and increased in..."

Response

We revise in text in Line 189-190 as “prevalence of rib and scpB had no change in serotype III and increase in serotypes V and VI between two studied years

Line 185: Should be better rephrased “in number from 55 to 7 was found in serotype V..."

Response

We change accordingly.

Materials and Methods

Line 203: “Totally" ,the "y" is missing also,source of strains should be added.

Response

We revised accordingly.

Line 206: “0.5% TBE may be “0.5X TBE”. It shall be changed throughout the manuscript.

Line 212: “pneumoniae" ,the “e" is missing

Response

We changed accordingly.

Line 229: What type of sequencing is referred? There are  no such results in results paragraph (if it is not my misunderstanding) ,so why to refer sequencing.

Why do authors refer “randomly selected different PCR products". Please clarify the method or exclude it at all,if the results are not in the results paragraph.

Response

We revised as “We randomly selected different PCR products of scp-1 and scp-2 amplified from three serotypes for sequencing.”

Line 233: “conserved"

Response

We change accordingly.

Line 234: Please change to “protein was analyzed within 8 strains... .which were of serotype.."

Response

Serotypes are listed in 4.5.

Line 241-242: There is no verb in the sentence. May be authors would like to say 'The amino acid sequences were compared/were aligned....with these of 19 strains.. ."

Response

We revised as “The amino acid sequences of 8 strains were aligned with those of 19 strains”

Line 243: “was performed"

Response

We revised accordingly.

We sincerely thanks for your valuable comments to improve our manuscript.